# *GNB3* c.825C>T (rs5443) Polymorphism and Risk of Acute Cardiovascular Events after Renal Allograft Transplant

**DOI:** 10.3390/ijms23179783

**Published:** 2022-08-29

**Authors:** Tobias Peitz, Birte Möhlendick, Winfried Siffert, Falko Markus Heinemann, Andreas Kribben, Ute Eisenberger, Justa Friebus-Kardash

**Affiliations:** 1Department of Nephrology, University Hospital Essen, University of Duisburg-Essen, Hufelandstr. 55, 45147 Essen, Germany; 2Institute of Pharmacogenetics, University Hospital Essen, University of Duisburg-Essen, 45147 Essen, Germany; 3Institute for Transfusion Medicine, Transplantation Diagnostics, University Hospital Essen, University of Duisburg-Essen, 45147 Essen, Germany

**Keywords:** *GNB3* c.825C>T polymorphism, renal transplantation, myocardial infarction, acute peripheral artery occlusive disease

## Abstract

The c.825C>T single-nucleotide polymorphism (rs5443) of the guanine nucleotide-binding protein subunit β3 (*GNB3*) results in increased intracellular signal transduction via G-proteins. The present study investigated the effect of the *GNB3* c.825C>T polymorphism on cardiovascular events among renal allograft recipients posttransplant. Our retrospective study involved 436 renal allograft recipients who were followed up for up to 8 years after transplant. The *GNB3* c.825C>T polymorphism was detected with restriction fragment length polymorphism (RFLP) polymerase chain reaction (PCR). The *GNB3* TT genotype was detected in 43 (10%) of 436 recipients. Death due to an acute cardiovascular event occurred more frequently among recipients with the TT genotype (4 [9%]) than among those with the CC/CT genotypes (7 [2%]; *p* = 0.003). The rates of myocardial infarction (MI)–free survival (*p* = 0.003) and acute peripheral artery occlusive disease (PAOD)–free survival (*p* = 0.004) were significantly lower among T-homozygous patients. A multivariate analysis showed that homozygous *GNB3* c.825C>T polymorphism exerted only a mild effect for the occurrence of myocardial infarction (relative risk, 2.2; *p* = 0.065) or acute PAOD (relative risk, 2.4; *p* = 0.05) after renal transplant. Our results suggest that the homozygous *GNB3* T allele exerts noticeable effects on the risk of MI and acute PAOD only in the presence of additional nonheritable risk factors.

## 1. Introduction

Heterotrimeric G-proteins composed of α-, β-, and γ-subunits are implicated in signal transduction via G-protein-coupled receptors that are expressed in a variety of tissues [1]. Neurotransmitters, hormones and other signaling factors serve as ligands for the G-protein-coupled receptors. The interaction of G-proteins with its G-protein-coupled receptors results in transmission of the signals of these ligands activating diverse downstream signal cascades in the cells and thereby changing the cell function [2]. In the inactive state of the receptor-activated G-proteins, which bound to the inner surface of the cell membrane, the Gα-subunit is tightly associated with the Gβ- and Gγ-subunits. Gβ- and Gγ-subunits are also tightly bound to each other and can be considered as one functional unit. Attaching of the agonist to its G-protein-coupled receptor allows the conformational change in the receptor which starts to act as a guanine nucleotide exchange factor. Consequently, the GDP (Guanosine diphosphate) will be exchanged by GTP (Guanosine triphosphate) on the Gα-subunit of the heterotrimeric G-protein leading to the activation of the Gα-subunit and its subsequent dissociation from the Gβ/γ-dimer. Then, both GTP-bound Gα-subunit and released Gβ/γ-subunit are capable to stimulate various downstream effector proteins in order to turn on different cellular signal pathways. Both stimulatory and inhibitory Gα-subunits, Gαs and Gαi, transmit their signals via the adenylyl cyclases. Gαs stimulates adenylyl cyclase resulting in the conversion of ATP into cyclic AMP. The second messenger cAMP (Adenosine 3′5′ Cyclic Monophosphate) activates cAMP-regulated proteins, such as protein kinase A and cyclic nucleotide-gated channels. Vice versa, the Gαi inhibits several adenylyl cyclases leading to reduced cAMP production. Gαq-proteins induce the β-isoforms of phospholipase C, which transforms phosphatidylinositol 4,5-bisphosphate into inositol trisphosphate and membrane-bound diacylglycerol. Inositol trisphosphate is needed to open the calcium channels on the endoplasmic reticulum membrane, and membrane-bound diacylglycerol activates protein kinase C. On the other hand, Gβ/γ-subunit regulates adenylyl cyclases, phospholipase Cβ, inwardly rectifying K+ channel and voltage-gated Ca^2+^ channels. In order to terminate the G-protein mediated signal cascade and to restore the G-protein-coupled receptor, the Gα-subunit hydrolyzes the GTP to GDP and reunites with the Gβ/γ-subunit.

The *GNB3* gene is one of the five genes that encode for the β-subunits of G-proteins. The c.825C>T polymorphism is a widely studied polymorphism of *GNB3* which causes an in-frame deletion of 123 nucleotides within exon 9 of the *GNB3* gene [3]. The in-frame deletion leads to alternative splicing of the *GNB3* gene, and this splicing is associated with the production of a truncated Gβ3 molecule (Gβ3s) that lacks 41 amino acids, a characteristic resulting in an alteration of the three-dimensional structure of the β-subunit [3]. Thus, subjects carrying the T-allele express both the truncated Gβ3s and wild-type Gβ3, and thereby exhibit a gain of function and subsequent enhanced activation of G-protein-induced signaling pathways [4]. The generation of Gβ3s related to the *GNB3* c.825C>T polymorphism has been linked to increased signal transduction via the G-protein-coupled receptors in diverse human cells, such as B and T lymphocytes, neutrophils, and fat cells [4,5,6,7]. The novel Gβ3s protein differs from Gβ3, but it exhibits functional activity and causes enhanced activation of pertussis toxin-sensitive G-proteins in 825T-allele carriers [4]. Increased binding of GTPγS to permeabilized Sf9 insect cells was detected in cells expressing the Gβ3s protein [4]. Vasoconstrictive hormones such as norepinephrine and angiotensin II are predominantly using and activating the pertussis toxin-sensitive G-protein heterotrimers. The pertussis toxin-sensitive G-proteins are involved in α2-adrenergic coronary vasoconstriction. Thus, in in vitro studies of α2-adrenoceptor mediated coronary vasoconstriction was enhanced in cells derived from 825T-allele carriers expressing the Gβ3s [8]. Moreover, an augmentation of chemotaxis potential of neutrophils which is mediated via β/γ-subunits released from pertussis toxin-sensitive Gα-subunits, was found in 825T-allele carriers. Stimulation with interleukin-8 which activates the CXC receptor coupled to the pertussis toxin-sensitive G-protein resulted in the increased chemotactic response of neutrophils obtained from individuals exhibiting the Gβ3s protein [9]. COS-7 cells transfected with the Gβ3s protein also revealed increased the chemotaxis activity compared to the cells following transfection with the wild-type Gβ3 [9].

The distribution of the c.825C>T polymorphism differs between races. The highest frequency (80–90%) occurs among black Africans, whereas an average of only 45% of white patients exhibit the T-allele [10]. The presence of the c.825C>T polymorphism in the *GNB3* gene has been associated with higher Na/H exchange activity that may be responsible for augmented salt retention [11]. 

Vasoconstrictor hormones, such as angiotensin II and endothelin, which have a strong impact on the pathogenesis of arterial hypertension, use G-protein-coupled receptors. Healthy carriers of the T-allele have reported increased responses to these vasoconstrictors and elevated arterial stiffness [12]. Hence, in recent case-control studies, the T-allele has been associated with higher blood pressure among healthy white subjects [13]. The T-allele frequency is also higher among hypertensive patients, and the polymorphism is, in general, associated with low renin hypertension [11,14,15]. Otherwise, the presence of lipolysis in cells is lower among T-allele carriers than among subjects with the CC genotype [7]. Therefore, increased insulin resistance, higher body mass index, and increased risk of obesity have been observed among c.825C>T polymorphism carriers in European study populations, and these findings could contribute to low renin hypertension among T-allele carriers [16,17]. Moreover, the prevalence of left ventricular hypertrophy is higher among hypertensive patients carrying the T-allele, probably because G-protein-mediated signaling is involved in the proliferation of fibroblasts in the left ventricle [16,18,19]. Several longitudinal trials have found that higher mortality and morbidity rates due to coronary artery diseases (CAD) are associated with the c.825C>T polymorphism [16]. 

However, the effect of the c.825C>T polymorphism among patients with impaired renal function has so far been poorly investigated. Blüthner et al. found that, among patients with diabetes mellitus type II (T2DM), those with the T-allele exhibit a predisposition for the development of end-stage renal disease (ESRD) [20]. On the other hand, a recently published study found that the outcome of renal transplantation is inferior when the renal allograft donors carry the TT genotype [18]. Although the donor’s genotype is also associated with a higher rate of chronic rejection, the recipient’s *GNB3* genotype exerts no influence on allograft survival or rejection events [21]. A study involving adult renal allograft recipients and another involving pediatric recipients found no relationship between the c.825C>T polymorphism and renal allograft survival or renal function after genotyping renal allograft recipients [22,23]. 

The current study was aimed at addressing the interaction between this specific *GNB3* polymorphism and cardiovascular mortality, cardiovascular events occurring after renal transplantation, and renal allograft outcome. It explored the effect of the T variant on the posttransplant occurrence of myocardial infarction (MI), acute peripheral artery occlusive disease (PAOD), stroke, posttransplant diabetes mellitus, rejection events, the appearance of de novo donor-specific anti-human leukocyte antibodies (anti-HLA DSAs), allograft loss, viral and bacterial infections, and cancers. 

## 2. Results

### 2.1. Patient Characteristics

The present study involved 436 renal allograft recipients, 183 (42%) women and 253 (58%) men, who underwent renal transplant at the University Hospital Essen between January 2011 and December 2015. Table 1 shows the characteristics of the study cohort. The median age was 53 years (range, 18 to 81 years). Cardiovascular pathologies existing before transplant (CAD, heart failure, cerebrovascular events, and chronic PAOD) were documented in 152 (35%) of the 436 patients. Overall, 93 (21%) recipients had CAD and 58 (13%) had chronic PAOD, which developed before renal transplant. 

Regarding the *GNB3* genotype distribution, the TT genotype was detected in 43 (10%) recipients, whereas the CT/CC genotypes were present in 393 (90%) recipients. 

It is of note that the incidence of preexisting CAD and chronic PAOD which occurred before transplant, while recipients were undergoing dialysis, did not vary between *GNB3* genotypes (Table 1). 

Pretransplant diabetes mellitus diagnosed before transplant during the time on dialysis was also similarly distributed between the TT and CT/CC genotype groups (Table 1). Stratification into T1DM and T2DM yielded no differences between the various *GNB3* genotypes (Table 1). Interestingly, carriers of the TT-allele were more likely than noncarriers to have diabetic nephropathy as a cause of chronic kidney disease (Table 1). In particular, the frequency of diabetic nephropathy due to T2DM was higher among TT genotype carriers than among carriers of the C-allele (Table 1), whereas the frequencies of diabetic nephropathy caused by T1DM were comparable between the genotype groups (Table 1). 

### 2.2. The TT Genotype of GNB3 Is Associated with a Higher Risk of Myocardial Infarction after Renal Transplant

We found no statistically significant difference between the *GNB3* genotype groups in the prevalence of CAD and chronic PAOD at the stage of end-stage renal disease requiring dialysis before transplant (Table 1). We investigated whether the presence of the TT genotype might be relevant for the development of acute MI after renal transplant. The median follow-up time of the study was 5.6 years and did not differ between the *GNB3* genotype groups (*p* = 0.86). A significantly higher frequency of death due to acute cardiovascular event after transplant was observed among TT genotype carriers (*p* = 0.003; Table 2). The survival rates were also significantly lower among TT genotype carriers (*p* = 0.003; Figure 1A). 

During the follow-up period after transplant, significantly more TT genotype carriers than CT/CC genotype carriers experienced acute MI (*p* = 0.007; Table 1). Additionally, the frequency of MI-free survival after renal transplant was lower among TT genotype carriers (*p* = 0.003; Figure 1B). 

We next performed a multivariate analysis examining the effect of the TT genotype on the occurrence of acute MI. We adjusted the analysis for several potential non-genetic covariables contributing to the risk of MI (Table 3). The previous univariate analysis had shown that most patients who experienced acute MI after transplant had diabetes mellitus before transplant. The results of that univariate analysis indicated that prolonged time on dialysis and older age were risk factors for the occurrence of MI after renal transplant (Table 3). A decrease in GFR of more than 50% of the baseline value reflecting progressive renal allograft dysfunction was not associated with a higher risk of MI after transplant (Table 3). Indeed, a multivariate analysis found that the risk of MI after transplant remained high among patients with preexisting diabetes mellitus and those who had undergone lengthy waiting times on dialysis before transplant; these factors contributed strongly to the development of acute MI (Table 3). However, when the multivariate analysis was adjusted to include relevant nongenetic covariates, the TT genotype exerted only a mild effect on the risk of MI (relative risk, 2.2; *p* = 0.065; Table 3) when compared to the other two relevant factors (time on dialysis and previous diabetes mellitus). 

### 2.3. The TT Genotype of GNB3 Is an Independent Risk Factor for the Occurrence of Acute PAOD after Renal Transplant

Subsequently, we assessed the association of the *GNB3* genotypes with the occurrence of acute PAOD after transplant. The numbers of recipients with pretransplant chronic PAOD were approximately similar among the TT genotype carriers and the CT/CC genotype carriers. As shown in Table 2, the incidence of acute PAOD after transplant was significantly higher among the TT genotype carriers than among the CT/CC genotype carriers (*p* = 0.006). Moreover, Kaplan-Meier survival plots illustrated the negative effect of the TT genotype on acute PAOD–free survival after transplant (*p* = 0.004; Figure 1C). 

A univariate analysis found that diabetes mellitus existing before renal transplant at the time of renal failure was likely to lead to the development of acute PAOD after transplant (odds ratio, 3.7) (Table 4). The results of the univariate analysis were confirmed by a multivariate Cox regression analysis (Table 4). The final multivariate analysis showed that the TT genotype of *GNB3* exerted a moderate effect on the risk of development of acute PAOD after transplant (relative risk, 2.4; *p* = 0.05; Table 4) when compared with pretransplant diabetes mellitus as an independent risk factor. 

### 2.4. The TT Genotype of GNB3 Is Not Associated with Allograft Failure or Rejection after Renal Transplant

With regard to allograft survival, we detected no *GNB3* genotype-dependent differences in the generation of de novo anti-HLA DSAs, the occurrence of different types of rejection, or the occurrence of bacterial or viral infection as important posttransplant complications (Table 2). The frequencies of occurrence of solid tumors, in particular squamous cell carcinoma, and of posttransplant lymphoproliferative disorders were also comparable between the two groups (Table 2).

## 3. Discussion

The main aim of the present study was to determine the effect of the *GNB3* c.825C>T polymorphism on cardiovascular morbidity and mortality and on renal allograft function after transplant. First, we observed a higher frequency of the TT genotype among recipients with T2DM among whom ESRD requiring dialysis later developed. In contrast, among patients with T1DM as the underlying cause of renal failure, the distribution of *GNB3* genotypes was equal. With respect to CAD and chronic PAOD occurring before transplantation among patients on dialysis, no association with *GNB3* genotypes was demonstrated. However, we found that recipients with the TT genotype were at higher risk of acute MI and acute PAOD after renal transplant than were recipients carrying the C allele. We detected no statistically significant influence of the *GNB3* c.825C>T polymorphism on allograft survival, rejection episodes, or the occurrence of de novo DSAs. 

Among the subset of recipients with T2DM we found a relationship between the TT genotype of the *GNB3* polymorphism and renal complications associated with that disease, such as diabetic nephropathy resulting in ESRD. In contrast, recipients with ESRD caused by T1DM exhibited comparable frequencies of the various *GNB3* genotypes. Our findings concerning patients with T2DM-associated nephropathy are in line with the findings of Blüthner et al., who found a higher frequency of the T-allele among patients with T2DM and renal damage than among patients who did not have diabetes [20]. Two other studies found no negative effects of *GNB3* c.825C>T polymorphism on renal function or the development of ESRD among patients with T2DM [24,25]. Our results concerning recipients with T1DM did not conflict with those of a study conducted by Fogarty et al. [26], who found that the *GNB3* c.825C>T polymorphism was an inappropriate marker of predisposing nephropathy among patients with this disease. 

Coronary arteriosclerosis proceeding CAD has a multifactorial origin and is a main risk factor for acute cardiovascular events, such as MI. Coronary artery calcification is partly determined by heritable factors. Single-nucleotide polymorphisms in the G-protein signaling pathways have been found to be associated with accelerated progression of CAD [16,27]. Several studies reported an association between the *GNB3* c.825C>T polymorphism and a higher risk of CAD or its severity [16,28,29,30,31]. The Heinz Nixdorf-Recall study, which involved 3108 participates, found that the presence of the T-allele was linked to rapid progression of calcification of coronary arteries, a finding implying the relevance of the *GNB3* c.825C>T polymorphism to genetic heritability of CAD [30]. In contrast, other case-control and longitudinal studies did not confirm a greater likelihood of cardiovascular mortality and morbidity among T-allele carriers in white populations [32,33,34,35]. The current study found that *GNB3* genotypes were equally distributed among patients with or without CAD or with or without chronic PAOD, as determined by medical history and diagnosed before transplant. The most important causative factor for the calcification of arterial vessels is arterial hypertension. The TT genotype of *GNB3* is well known to be associated with arterial hypertension. Notably, the effects of the *GNB3* c.825C>T polymorphism on arterial hypertension among T-allele carriers are restricted to low renin levels [11]. Almost all patients in our study cohort exhibited secondary arterial hypertension associated with renal dysfunction, as well as the triggering activation of the renin-angiotensin-system, resulting in high levels of renin among these selected patients with end-stage chronic kidney disease. Consequently, the presence of high renin arterial hypertension, which was found among most of the recipients in our cohort, may explain the lack of differences in the occurrence of CAD and PAOD between polymorphism carriers and noncarriers among patients undergoing dialysis. 

Interestingly, death due to acute cardiovascular events, acute MI, or acute PAOD after renal transplant was more frequent among patients with the TT genotype than among patients with the CT/CC genotypes. These results agree with those of previous studies showing that the TT genotype is a risk factor for MI and stroke in several populations [16,29]. For the first time, we examined the relationship between *GNB3* genotypes and MI or acute PAOD in a subset of renal transplant patients. After transplant, the majority of the recipients experienced a relevant improvement in renal function, a finding suggesting the reduction of renin overexpression. In this fashion, heritable factors, such as the *GNB3* polymorphism, seem to exert stronger effects on renal allograft recipients after transplant than before, when the recipients were undergoing dialysis. However, multivariate analysis showed that the association between the *GNB3* polymorphism and the risk of MI or acute PAOD was not statistically significant but was close to the level of significance (MI, *p* = 0.086; PAOD, *p* = 0.05). It is conceivable that the *GNB3* polymorphism in general exerts a cumulative effect on the progression of CAD and the risk of acute cardiovascular events, such as MI and acute PAOD, along with a number of other nonheritable factors. We added the variable of diabetes mellitus existing before transplant to our multivariate analysis; this condition is a pivotal factor predisposing patients to MI or acute PAOD and strongly affecting the relative risk of acute events. Indeed, the isolated influence of the *GNB3* polymorphism on cardiovascular diseases appears to be mild. 

In the present study, we focused on the relevance of the c.825C>T (rs5443) polymorphism for the risk of occurrence of cardiovascular events after renal transplant, because this polymorphism represents the best described and most studied variant with functional consequences in the gene *GNB3*. Up until now, over 700 studies investigated the impact of this polymorphism on arterial hypertension, obesity, cardiovascular events, and many other disorders. Nevertheless, in the last years several reports on other polymorphisms identified in the intron or exon regions of the *GNB3* were published. The research group of Rosskopf et al. described three additional polymorphisms of the *GNB3* gene, c.814G>A (rs5442), c.657A>T (rs45476395) and c.1429C>T (rs5446) [36]. While these three polymorphisms were associated with obesity, no association with essential arterial hypertension was detected [37,38]. Another single-nucleotide polymorphism of *GNB3*, the rs2301339, was linked to the young-onset hypertension in the Chinese population considering 992 young-onset hypertensive cases and 992 matched controls [39]. Furthermore, increased fasting plasma glucose levels were detected in Japanese patients suffering from diabetes mellitus type II and having the rs2301339 polymorphism of *GNB3* [40]. However, in comparison with the c.825C>T (rs5443) polymorphism of *GNB3*, the number of reports on the additional polymorphisms of *GNB3* gene is low und the evidence of their contribution to the arterial hypertension, obesity and cardiovascular events is weak. 

Furthermore, there are five human Gβ-subunit proteins which are encoded by five different *GNB* genes, *GNB1*, *GNB2*, *GNB3*, *GNB4*, and *GNB5*, and are highly similar. Polymorphisms or mutations in *GNB1*, *GNB2*, and *GNB5* genes were reported to be associated with neurodevelopmental disorders [41,42,43]. For the variants of *GNB4* an association with Charcot-Marie-Tooth disease and refractive disorders was described [44,45]. However, data on the relationship of polymorphisms in *GNB1*, *GNB2*, *GNB4*, and *GNB5* with cardiovascular events, in particular, related to renal allograft recipients are lacking.

Next, we assessed the effect of the *GNB3* c.825C>T polymorphism on the renal allograft outcome after transplant. Homozygous T-allele carriers are prone to magnified chemotaxis and migration of neutrophils in response to stimulation with interleukin-8 in in vitro experiments [9]. A genotype-dependent effect on the proliferation of CD4^+^ T cells after in vitro administration of interleukin-2 has also been pointed out by Lindemann et al. [6,46]. Hence, we postulated that the *GNB3* c.825C>T polymorphism may be a promising candidate negatively influencing allograft survival and promoting rejection. A study by Beige et al. involving 320 renal transplant patients found that the *GNB3* genotype of the donor rather than of the recipient is apparently of primary relevance for allograft survival and occurrence of rejection events [21]. The authors hypothesized that the kidneys of donors with the TT genotype characterized by overexpression and activation of stimulatory G-protein may be prone to increased T-cell response and elevated risk of arterial hypertension lesions in renal allografts, changes that lead to accelerated allograft injury and chronic rejection. Two other clinical trials found that recipient genotype has no effect on the allograft outcome in adult and pediatric renal allograft recipients [22,23]. Our data also failed to demonstrate any influence of the *GNB3* c.825C>T polymorphism among renal allograft recipients on the frequency of rejection events, the development of de novo DSA, or allograft loss, findings supporting previous observations. Regrettably, information about donor genotype was not available in the present study.

The current cohort of renal transplant recipients was of medium size, and the mean duration of cardiovascular disease after transplantation was 8 years. Nevertheless, the number of cardiovascular events was low, a finding reflecting the low power and the imposing limitations of the study. We considered the lack of donor genotype to be an additional limitation. Although the TT genotype of the *GNB3* c.825C>T polymorphism exerted a mild effect on the risk of acute MI and acute PAOD after renal transplant, it could be regarded as a potential genetic marker helping to define the cardiovascular risk profile after renal transplant (Figure 2). Homozygous T-allele carriers among renal allograft recipients may benefit from close clinical controls and from a strict lifestyle modification and treatment aimed at decreasing the risk of acute cardiovascular events. Furthermore, prospective trials in particular are needed to explore the effect of the TT genotype of the *GNB3* c.825C>T polymorphism on posttransplant prognosis for renal allograft recipients. 

## 4. Materials and Methods

### 4.1. Study Population

The study cohort consisted of 436 renal allograft recipients who were admitted to the University Hospital Essen for transplant from January 2011 to December 2015. The current single-center study retrospectively analyzed the association between the *GNB3* c.825C>T polymorphism and clinical outcome variables after renal transplant, including allograft survival, appearance of de novo DSAs, and rejection events, as well as the occurrence of MI and acute pretransplant PAOD. Exclusion criteria were aged less than 18 years and allograft loss within the first 90 days after transplant. The study protocol was approved by the Ethics Committee of the University Hospital Essen (19–9071-BO).

Peripheral blood was collected once before transplant when patients underwent dialysis. Clinical and laboratory data were obtained by a review of medical records. Renal allograft recipients were followed up for as long as 10 years after transplant. Acute MI and acute PAOD events after renal transplantation were obtained from medical history. The diagnosis of acute MI was based on typical electrocardiographic changes and an increase in the activity of cardiac enzymes. Only acute PAOD that required arterial revascularization, by either open surgery or an endovascular procedure, was included in the analysis. Pretransplant cardiovascular morbidity was defined as CAD, heart failure, or MI, as well as cerebrovascular events, stroke, or chronic PAOD occurring before transplant. Posttransplant diabetes mellitus was defined as new-onset diabetes mellitus after renal transplant, corresponding to the recommendations of the Deutsche Diabetes-Gesellschaft and the American Diabetes Association (ADA) [47].

All documented rejection episodes were biopsy-proven. Biopsies were performed only for cause during the study period, and samples were analyzed according to the latest available Banff grading criteria [48]. Experienced nephropathologists examined all renal transplant specimens with light microscopy and immunohistochemical analyses. The estimated glomerular filtration rate (eGFR) was calculated with the Chronic Kidney Disease Epidemiology Collaboration (CKD-EPI) equation [49]. Allograft failure was defined as a return to dialysis. GFR reduction was defined as a reduction in renal function of more than 50% during the 8-year follow-up period.

Cytomegalovirus (CMV) infection was determined by CMV viremia of more than 65 IU/mL. BK polyomavirus (BKV) viremia was characterized by BKV DNA of more than 400 copies per mL. The diagnosis of BKV nephropathy was proven by renal biopsy. Epstein-Barr virus (EBV) reactivation was suspected when EBV viremia of more than 1000 IU/mL was detected. 

Table 1 summarizes the demographic, clinical, and laboratory data of the study cohort. Preformed anti-HLA antibodies were detected in 163 (37%) recipients. Most patients underwent induction therapy with basiliximab; thymoglobulin was used as induction therapy only in case of the presence of more than 25% of panel-reactive antibody (PRA). ABO-incompatible transplant required pretreatment with a single intravenous dose of 500 mg rituximab, immunoadsorption, and intravenous immunoglobulin. The standard maintenance immunosuppressive regimen consisted of tacrolimus or cyclosporine A, mycophenolate mofetil (MMF) or mycophenolic acid (MPA), and steroids. Most recipients were treated with tacrolimus administered twice daily; 26 patients received an extended-release formulation of tacrolimus administered once daily; 34 patients were treated with cyclosporine A; and 73 patients were treated with mammalian target of rapamycin (mTOR) inhibitors, such as everolimus or sirolimus.

### 4.2. HLA Typing of Recipients and Donors

For HLA typing of recipients and donors, we isolated DNA from peripheral blood samples. HLA class I (HLA-A, -B, -C) and class II (HLA-DRB1, -DQB1) typing was performed at the first-field resolution level, as previously described [50]. Second-field typing was performed to type for selected high-resolution HLA alleles and serological equivalents, according to established Eurotransplant procedures [51]. HLA-DP and HLA-DQA typing was not performed, and HLA-DP-and HLA-DQA-specific antibodies were excluded from further analysis with respect to the putative donor specificity of the anti–HLA DP and DQA antibodies.

### 4.3. HLA Antibody Detection and Specification

All patients were screened for anti-HLA class I and II antibodies before transplant. The pretransplant patient sera collected closest to the date of transplant were used for screening. Pretransplant sensitization status was determined for all patients with the standard immunoglobulin G (IgG) complement-dependent cytotoxicity (CDC) test with and without the addition of dithiothreitol (DTT) to exclude antibodies of the IgM isotype. In addition, all patients were tested with a Luminex-based LABScreen™ Mixed bead assay (One Lambda, Thermo Fisher Scientific Inc., Waltham, MA, USA). In the step-by-step analysis [52], the anti-HLA class I or II positive sera were subsequently specified with LABScreen™ single-antigen bead (SAB) assays (One Lambda, Thermo Fisher Scientific Inc., Waltham, MA, USA). All beads with normalized median fluorescence intensity (MFI) values higher than 1000 were considered to be positive for anti-HLA antibodies. To address the potential effect of interfering antibodies or prozone effects on our MFI analyses, we analyzed the sera after multiple freezing and thawing cycles and treatment with ethylenediaminetetraacetic acid (EDTA) [53]. 

The results of pretransplant lymphocytotoxic T-cell crossmatches (CDC crossmatch) were negative for all recipients. The anti-HLA antibody status after transplant was monitored at months 3, 6, and 12 after transplant and annually thereafter. Additional screening was performed in case of allograft dysfunction. For the current study, de novo anti-HLA antibodies were detected as early as 4 weeks after renal transplant. We considered samples to be positive for de novo anti-HLA antibodies only when the antibodies were detected at least twice. Nonrecurring evidence of weak anti-HLA antibodies after transplant was considered to be an artifact and was therefore not considered. 

### 4.4. GNB3 rs5443 Genotyping

Genomic DNA was extracted from 200 µL EDTA-treated blood with the QIAamp DNA Blood Mini Kit (Qiagen, Hilden, Germany). The polymerase chain reaction (PCR) was performed with 2 µL genomic DNA and 30 µL Taq DNA-Polymerase 2× Master Mix Red (Ampliqon, Odense, Denmark) with the following conditions: initial denaturation at 94 °C for 3 min; 38 cycles with denaturation at 94 °C for 30 s, annealing at 60 °C for 30 s, and elongation at 72 °C for 30 s each; final elongation at 72 °C for 10 min (forward primer: 5′GCCCTCAGTTCTTCCCCAAT3′; reverse primer 3′CCCACACGCTCAGACTTCAT5′). PCR products were digested with BseDI (Thermo Scientific, Dreieich, Germany), and restriction fragments were analyzed by agarose gel electrophoresis. For the various genotypes, the results of restriction fragment length polymorphism (RFLP)-PCR were validated by Sanger sequencing. The Hardy–Weinberg equilibrium (HWE) was calculated with Pearson’s chi square (*χ*^2^) goodness-of-fit test, and samples were considered to be deviant from HWE at a significance level of *p* ≤ 0.05. Genotypes for *GNB3* rs5443 were compatible with HWE (*χ*^2^ = 0.03; *p* = 0.87).

### 4.5. Statistical Analysis

The categorical variables were expressed as numbers and percentages and were compared with the two-tailed *χ^2^* test. The significant differences between continuous variables were determined with Student’s *t*-test. The comparisons of survival rates were performed with the log-rank test. The multivariate Cox regression analysis was used to explore whether the association between *GNB3* polymorphism and acute MI or acute PAOD was independent of other covariates. Statistical significance was set at the level of *p* ≤ 0.05. The statistical analyses were calculated with GraphPad Prism version 6 (GraphPad Software, Inc., La Jolla, CA, USA) and IBM SPSS Statistics version 23 (IBM Corp., Armonk, NY, USA).

## Figures and Tables

**Figure 1 ijms-23-09783-f001:**
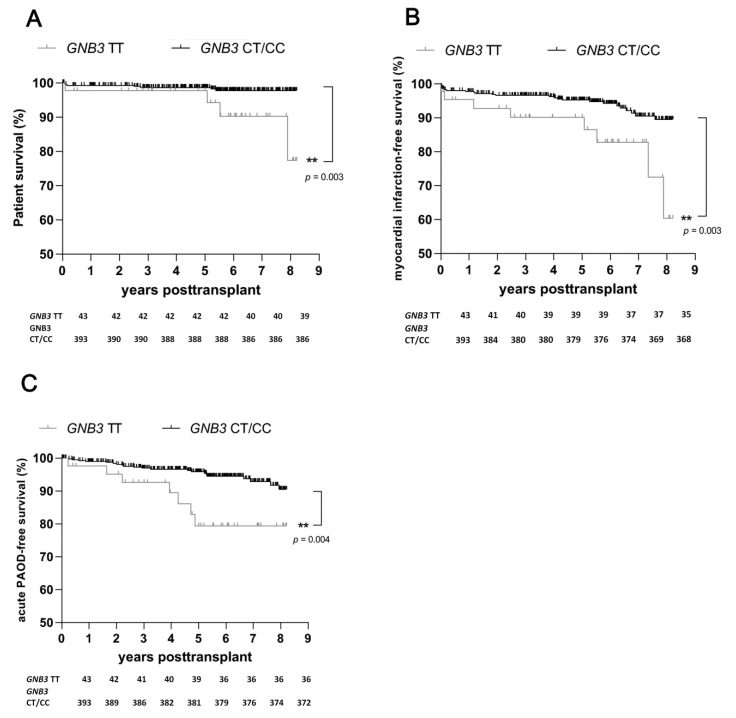
Occurrence of acute cardiovascular events among carriers of TT genotype and carriers of the CT/CC genotype of *GNB3* during an 8-year follow-up period after transplant. (**A**) Death due to cardiovascular event according to the genotype of *GNB3* (*p* = 0.003). (**B**) Survival dependent on the occurrence of acute myocardial infarction after transplant among carriers of the TT or the CT/CC genotype of *GNB3* (*p* = 0.003). (**C**) Survival dependent on the development of acute peripheral artery occlusive disease (PAOD) after transplant among carriers of the TT or the CT/CC genotype of *GNB3* (*p* = 0.004). **, *p* = 0.01. PAOD, peripheral artery occlusive disease.

**Figure 2 ijms-23-09783-f002:**
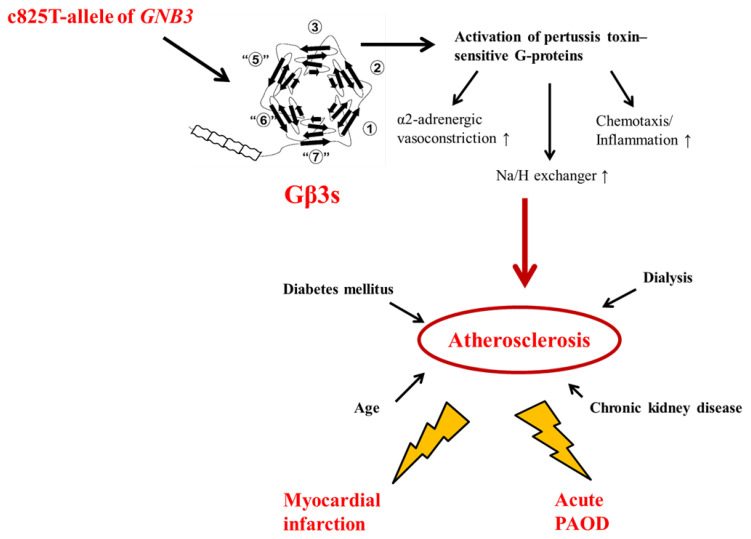
Impact of the *GNB3* TT genotype on occurrence of acute cardiovascular events such as myocardial infarction and acute PAOD after renal allograft transplant. The 825T-allele of *GNB3* leads to production of a truncated Gβ3 molecule (Gβ3s). The Gβ3s induces increased activation of pertussis toxin-sensitive G-proteins that lead to enhanced α2-adrenergic vasoconstriction, neutrophil chemotaxis response and subsequent potential activation of Na/H exchanger. Thus, the Gβ3s might contribute to the accelerated development of atherosclerosis in renal allograft recipients. Besides of the presence of the TT genotype of *GNB3*, other nonheritable factors such as age, chronic kidney disease, required dialysis treatment and concomitant occurrence of diabetes mellitus play an important role in formation of atherosclerosis in renal allograft recipients from our cohort. In combination with these above mentioned nonheritable factors, the presence of TT-genotype of *GNB3* is associated with increased risk for acute myocardial infarction and PAOD after renal transplant.

**Table 1 ijms-23-09783-t001:** Baseline characteristics of 436 renal allograft recipients.

	All Patients n = 436	*GNB3* TT n = 43	*GNB3* CC/CT n = 393	χ^2^	OR (CI 95%)	*p* Value
Recipients						
Age in years, median (range)	53 (18–81)	56 (18–71)	53 (18–81)			0.27
Women, n (%)	183 (42)	13 (30)	170 (43)	2.70	0.57 (0.28–1.1)	0.10
Previous transplants, n (%)	52 (12)	5 (12)	47 (12)	0.004	0.97 (0.4–2.51)	0.95
CMV status positive, n (%)	270 (62)	30 (70)	240 (61)	1.24	1.47 (0.76–2.99)	0.26
CMV high risk (D+/R−), n (%)	75 (17)	5 (12)	70 (18)	1.04	0.61 (0.25–1.52)	0.31
PRA, n (%)	37 (8)	3 (7)	34 (9)	0.14	0.79 (0.25–2.56)	0.71
Preformed anti-HLA antibodies, n (%)	163 (37)	15 (35	148 (38)	0.13	0.89 (0.47–1.69)	0.72
Class I, n (%)	89 (20)	7 (16)	82 (21)	0.50	0.74 (0.3–1.68)	0.48
Class II, n (%)	27 (6)	4 (9)	23 (6)	0.79	1.65 (0.59–4.91)	0.37
Class I and II, n (%)	47 (11)	4 (9)	43 (11)	0.11	0.83 (0.31–2.25)	0.74
Preformed anti-HLA DSA, n (%)	38 (9)	3 (7)	35 (9)	0.18	0.77 (0.24–2.47)	0.67
Rest diuresis in ml, median (range)	500 (0–3000)	500 (0–2500)	500 (0–3000)			0.36
Delayed graft function, n (%)	99 (23)	9 (21)	90 (23)	0.09	0.89 (0.41–1.85)	0.77
Cold ischemia time in minutes, median (range)	660 (58–1869)	507 (98–1177)	665 (58–1869)			0.84
Warm ischemia time in minutes, median (range)	26 (11–82)	27 (11–50)	26 (11–82)			0.85
Coronary artery disease pretransplant, n (%)	93 (21)	12 (28)	81 (21)	1.23	1.49 (0.75–3.03)	0.27
Chronic PAOD, n (%)	58 (13)	9 (21)	49 (12)	2.41	1.86 (0.82–4.1)	0.12
Diabetes mellitus pretransplant, n (%)	68 (16)	9 (21)	59 (15)	1.03	1.50 (0.67–3.21)	0.31
Diabetes mellitus type I pretransplant, n (%)	18 (4)	3 (7)	15 (4)	0.98	1.89 (0.56–6.64)	0.32
Diabetes mellitus type II pretransplant, n (%)	48 (11)	5 (12)	43 (11)	0.02	1.07 (0.44–2.8)	0.89
Diabetes mellitus type III pretransplant, n (%)	2 (1)	1 (2)	1 (1)	3.64	9.33 (0.48–177.6)	0.06
Donors						
Deceased, n (%)	313 (72)	27 (63)	286 (73)	1.91	0.63 (0.33–1.26)	0.17
Age in years, median (range)	52 (0–82)	52 (22–76)	52 (0–82)			0.58
Female, n (%)	203 (47)	21 (49)	182 (46)	0.10	1.11 (0.58–2.09)	0.75
CMV status, +/−, n (%)	245 (56)	24 (56)	221 (56)	0.003	0.98 (0.53–1.8)	0.96
ABO-incompatible transplant, n (%)	33 (8)	6 (14)	27 (7)	2.78	2.20 (0.88–5.72)	0.10
Immunosuppression at time of transplant						
IL-2 receptor antagonist, n (%)	405 (93)	36 (84)	369 (94)	6.07	0.33 (0.13–0.87)	0.01
ATG, n (%)	23 (5)	5 (12)	18 (5)	3.85	2.74 (1.06–7.86)	0.05
Calcineurin inhibitor, n (%)	436 (100)	43 (100)	393 (100)			
Tacrolimus, n (%)	402 (92)	41 (95)	361 (92)	0.66	1.82 (0.48–7.94)	0.42
Tacrolimus extended-release formulation, n (%)	26 (6)	5 (12)	21 (5)	2.73	2.33 (0.91–6.39)	0.10
Cyclosporine A, n (%)	34 (8)	2 (5)	32 (8)	0.66	0.55 (0.13–2.08)	0.42
mTOR inhibitor, n (%)	73 (17)	7 (16)	66 (17)	0.007	0.96 (0.39–2.23)	0.93
MMF/MPA, n (%)	362 (83)	36 (84)	326 (83)	0.02	1.06 (0.46–2.6)	0.90
Steroids, n (%)	436 (100)	43 (100)	393 (100)			
Rituximab, n (%)	5 (1)	2 (5)	3 (1)	5.17	6.34 (1.09–31.54)	0.02
Other, n (%)	3 (1)	0	3 (1)	0.33	0 (0.0–10.64)	0.57
HLA mismatches						
MM (A/B), n (%)	362 (83)	35 (81)	327 (83)	0.09	0.88 (0.4–1.89)	0.76
HLA class I MM (A/B): 1–2, n (%)	214 (49)	16 (37)	198 (50)	2.69	0.58 (0.3–1.1)	0.10
HLA class I MM (A/B): 3–4, n (%)	148 (34)	19 (44)	129 (33)	2.23	1.62 (0.88–3.05)	0.14
MM (DR), n (%)	312 (72)	28 (65)	284 (72)	0.97	0.72 (0.37–1.36)	0.32
HLA class II MM (DR): 1, n (%)	206 (47)	16 (37)	190 (48)	1.93	0.63 (0.32–1.19)	0.16
HLA class II MM (DR): 2, n (%)	106 (24)	12 (28)	94 (24)	0.34	1.23 (0.63–2.49)	0.56
Causes of renal failure						
1. Diabetic glomerulosclerosis, n (%)	41 (9)	8 (19)	33 (8)	4.74	2.49 (1.13–5.6)	0.03
due to diabetes mellitus type II, n (%)	23 (5)	5 (12)	18 (5)	3.85	2.74 (1.06–7.86)	0.05
due to diabetes mellitus type I, n (%)	18 (4)	3 (7)	15 (4)	0.98	1.89 (0.56–6.64)	0.32
2. Chronic glomerulonephritis, n (%)	117 (27)	10 (23)	107 (27)	0.31	0.81 (0.4–1.71)	0.58
3. Nephrosclerosis, n (%)	58 (13)	8 (19)	50 (13)	1.16	1.57 (0.73–3.55)	0.28
4. Polycystic kidney disease, n (%)	66 (15)	5 (12)	61 (15)	0.46	0.72 (0.3–1.81)	0.50
5. Tubulointerstitial nephritis, n (%)	15 (3)	0	15 (4)	1.70	0 (0.0–2.28)	0.19
6. Congenital anomalies, n (%)	39 (9)	6 (14)	33 (8)	1.47	1.77 (0.73–4.43)	0.23
7. Autoimmune disease, n (%)	18 (4)	0	18 (5)	2.05	0 (0.0–1.82)	0.15
8. Amyloidosis, n (%)	4 (1)	0	4 (1)	0.44	0 (0.0–9.44)	0.51
9. Reflux nephropathy/recurrent pyelonephritis, n (%)	21 (5)	3 (7)	18 (5)	0.49	1.56 (0.47–5.19)	0.49
10. HUS, n (%)	8 (2)	0	8 (2)	0.89	0 (0.0–4.23)	0.35
11. Other, n (%)	49 (11)	4 (9)	45 (12)	0.18	0.79 (0.29–2.13)	0.67

Anti-HLA, anti–human leukocyte antigen; ATG, anti-thymocyte globulin; CMV, cytomegalovirus; D, donor; DSA, donor-specific antibody; HLA, human leukocyte antigen; HUS, hemolytic uremic syndrome; IL-2, interleukin-2; MM, mismatch; MMF, mycophenolate mofetil; MPA, mycophenolic acid; mTOR, mammalian target of rapamycin; OR, odds ratio; PAOD, peripheral artery occlusive disease; PRA, panel-reactive antibodies; R, recipient.

**Table 2 ijms-23-09783-t002:** Characteristics of acute cardiovascular events after transplant, as well as renal allograft outcome, infectious complications, and occurrence of malignant tumors after renal transplant among carriers of the TT genotype or the CT/CC genotype of *GNB3*.

	All Patients n = 436	*GNB3* TT n = 43	*GNB3* CC/CT n = 393	χ^2^	OR (CI 95%)	*p* Value
Acute cardiovascular events after transplant						
Death due to cardiovascular event, n (%)	11 (3)	4 (9)	7 (2)	8.92	5.66 (1.78–18.83)	0.003
Myocardial infarction, n (%)	35 (8)	8 (19)	27 (7)	7.23	3.1 (1.37–7.23)	0.007
Stroke, n (%)	13 (3)	1 (2)	12 (3)	0.07	0.76 (0.07–4.32)	0.79
Acute PAOD, n (%)	28 (6)	7 (16)	21 (5)	7.71	3.44 (1.33–8.24)	0.006
Posttransplant diabetes mellitus, n (%)	75 (17)	3 (7)	72 (18)	3.50	0.33 (0.11–1.01)	0.06
Allograft outcome						
Cellular rejection, n (%)	81 (19)	7 (16)	74 (19)	0.17	0.84 (0.34–1.93)	0.68
Borderline rejection, n (%)	77 (18)	2 (5)	75 (19)	5.55	0.21 (0.05–0.8)	0.02
Cellular and borderline rejection, n (%)	137 (31)	9 (21)	128 (33)	2.44	0.55 (0.25–1.18)	0.12
ABMR with DSAs, n (%)	28 (6)	0 (0)	28 (7)	3.27	0 (0.0–1.07)	0.07
All ABMR, n (%)	56 (13)	2 (5)	54 (14)	2.86	0.31 (0.07–1.21)	0.09
All rejections, n (%)	155 (36)	10 (23)	145 (37)	3.15	0.52 (0.26–1.08)	0.08
Multiple cellular/borderline rejections, n (%)	29 (7)	1 (2)	28 (7)	1.44	0.31 (0.03–1.73)	0.23
Transplant failure, n (%)	52 (12)	5 (12)	47 (12)	0.004	0.97 (0.4–2.51)	0.95
Decrease in eGFR, n (%)	84 (19)	6 (14)	78 (20)	0.87	0.65 (0.28–1.6)	0.35
de novo anti-HLA antibodies, n (%)	117 (27)	13 (30)	104 (26)	0.28	1.2 (0.59–2.38)	0.60
Class I, n (%)	42 (10)	4 (9)	38 (10)	0.006	0.96 (0.35–2.62)	0.94
Class II, n (%)	42 (10)	6 (14)	36 (9)	1.02	1.61 (0.66–3.97)	0.31
Class I and II, n (%)	33 (8)	3 (7)	30 (8)	0.02	0.91 (0.28–2.99)	0.88
de novo anti HLA DSA, n (%)	51 (12)	4 (9)	47 (12)	0.26	0.76 (0.28–2.02)	0.61
Class I, n (%)	17 (4)	1 (2)	16 (4)	0.32	0.56 (0.05–3.47)	0.57
Class II, n (%)	24 (6)	2 (5)	22 (6)	0.07	0.82 (0.19–3.31)	0.80
Class I and II, n (%)	10 (2)	1 (2)	9 (2)	0.0002	1.01 (0.09–6.49)	0.99
Infections						
CMV infection, n (%)	162 (37)	11 (26)	151 (38)	2.74	0.55 (0.27–1.12)	0.10
CMV disease, n (%)	34 (8)	4 (9)	30 (8)	0.15	1.24 (0.45–3.51)	0.70
BKV viremia, n (%)	101 (23)	13 (30)	88 (22)	1.34	1.5 (0.72–2.99)	0.25
BKV nephropathy, n (%)	30 (7)	3 (7)	27 (7)	0.001	1.02 (0.31–3.1)	0.98
HEV infection, n (%)	11 (3)	1 (2)	10 (3)	0.008	0.91 (0.08–5.56)	0.93
EBV reactivation, n (%)	84 (19)	11 (26)	73 (19)	1.22	1.51 (0.71–3.16)	0.27
Influenza A and B infections, n (%)	19 (4)	0 (0)	19 (5)	2.17	0 (0.0–1.7)	0.14
Norovirus infection, n (%)	9 (2)	2 (5)	7 (2)	1.58	2.69 (0.55–12.76)	0.21
HSV infection, n (%)	6 (1)	0 (0)	6 (2)	0.67	0 (0.0–6.65)	0.41
VZV/Zoster infection, n (%)	11 (3)	0 (0)	11 (3)	1.24	0 (0.0–3.43)	0.27
Pyelonephritis, n (%)	122 (28)	10 (23)	112 (28)	0.53	0.76 (0.37–1.6)	0.47
More than 1 episode, n (%)	64 (15)	4 (9)	60 (15)	1.10	0.57 (0.21–1.61)	0.29
Pneumonia, n (%)	62 (14)	6 (14)	56 (14)	0.003	0.98 (0.42–2.29)	0.96
More than 1 episode, n (%)	21 (5)	2 (5)	19 (5)	0.003	0.96 (0.21–4.0)	0.96
Sepsis, n (%)	85 (19)	7 (16)	78 (20)	0.31	0.79 (0.32–1.8)	0.58
More than 1 episode, n (%)	21 (5)	3 (7)	18 (5)	0.49	1.56 (0.47–5.19)	0.49
Malignant tumors						
Solid malignant tumor, n (%)	47 (11)	6 (14)	41 (10)	0.50	1.39 (0.58–3.38)	0.48
Squamous cell carcinoma, n (%)	52 (12)	4 (9)	48 (12)	0.31	0.74 (0.27–1.97)	0.58
PTLD, n (%)	2 (1)	0 (0)	2 (1)	0.22	0 (0.0–19.89)	0.64

ABMR, antibody-mediated rejection; anti-HLA, anti–human leukocyte antigen; BKV, BK virus; CMV, cytomegalovirus; DSA, donor-specific antibody; EBV, Epstein-Barr virus; eGFR, estimated glomerular filtration rate; HEV, hepatitis E virus; HLA, human leukocyte antigen; HSV, herpes-simplex virus; OR, odds ratio; PAOD, peripheral artery occlusive disease; PTLD, post-transplant lymphoproliferative disorder; VZV, varicella-zoster virus.

**Table 3 ijms-23-09783-t003:** Results of univariate and multivariate analyses identifying risk factors for the development of acute myocardial infarction among 436 patients after renal allograft transplant.

	Myocardial Infarction n = 35	Patients without Myocardial Infarction n = 401	Univariate Relative Risk (95% CI)	*p* Value	Multivariate Relative Risk (95% CI)	*p* Value
Variable						
Age in years, median (range)	59 (29–71)	52 (18–81)		0.005	1.04 (1.01–1.08)	0.01
Women, n (%)	11 (31)	172 (43)	0.73 (0.43–1.14)	0.19		
Diabetes mellitus pretransplant, n (%)	11 (31)	57 (14)	2.21 (1.24–3.63)	0.007	3.72 (1.71–8.09)	<0.001
Nephrosclerosis as cause of ESRD, n (%)	5 (14)	53 (13)	1.08 (0.46–2.33)	0.86		
*GNB3* TT genotype, n (%)	8 (23)	35 (9)	2.62 (1.29–4.93)	0.007	2.23 (0.95–5.24)	0.065
Follow-up time, median (range)	2113 (379–3474)	2027 (6–3756)		0.35		
Time on dialysis, median (range)	1445 (11–6822)	1078 (0–12379)		0.02	1.00 (1.00–1.00)	<0.001
IL-2 receptor antagonist, n (%)	33 (94)	372 (93)	1.02 (0.88–1.08)	0.74		
ATG, n (%)	2 (6)	21 (5)	1.09 (0.29–3.82)	0.9		
Decrease in eGFR, n (%)	7 (20)	77 (19)	1.04 (0.51–1.94)	0.91		

ATG, anti-thymocyte globulin; CI, confidence interval; eGFR, estimated glomerular filtration rate; ESRD, end-stage renal disease; IL-2, interleukin-2.

**Table 4 ijms-23-09783-t004:** Results of univariate and multivariate analyses identifying risk factors for the development of acute peripheral artery occlusive disease among 436 patients after renal allograft transplant.

	Acute PAOD n = 28	Patients without Acute PAOD n = 408	Univariate Relative Risk (95% CI)	*p* Value	Multivariate Relative Risk (95% CI)	*p* Value
Variable						
Age in years, median (range)	53 (32–70)	53 (18–81)		0.64		
Women, n (%)	9 (32)	174 (43)	0.75 (0.42–1.21)	0.28		
Diabetes mellitus pretransplant, n (%)	13 (46)	55 (13)	3.44 (2.07–5.25)	<0.001	5.55 (2.59–11.91)	<0.001
Nephrosclerosis as cause of ESRD, n (%)	4 (14)	54 (13)	1.08 (0.42–2.49)	0.87		
*GNB3* TT genotype, n (%)	7 (25)	36 (9)	2.83 (1.35–5.40)	0.006	2.39 (1.00–5.72)	0.05
Follow-up time in days, median (range)	2154 (1153–3418)	2027 (6–3756)		0.12		
Time on dialysis, median (range)	1321 (152–5895)	1109 (0–12379)		0.19		
IL-2 receptor antagonist, n (%)	24 (86)	381 (93)	0.92 (0.73–1.02)	0.13		
ATG, n (%)	2 (7)	21 (5)	1.39 (0.37–4.76)	0.65		
Decrease in eGFR, n (%)	4 (14)	80 (20)	0.73 (0.29–1.65)	0.49		

ATG, anti-thymocyte globulin; CI, confidence interval; eGFR, estimated glomerular filtration rate; ESRD, end-stage renal disease; IL-2, interleukin-2; PAOD, peripheral artery occlusive disease.

## Data Availability

The data presented in this study are available on request from the corresponding author. The data are not publicly available due to restrictions e.g., privacy or ethical.

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
