# Peer review of "GNB3 c.825C>T (rs5443) Polymorphism and Risk of Acute Cardiovascular Events after Renal Allograft Transplant"

_ijms, 2022, doi:10.3390/ijms23179783_

Round 1
Reviewer 1 Report
The authors did a retrospective study involving 436 renal allograft recipients, and found that the patients with homozygous GNB3 T allele have lower myocardial infarction and acute peripheral artery occlusive
disease free survival rate.
Overall, the results are generally well presented, but a few issues will need to be addressed within the revised text of the manuscript before the paper can be published.
1, in table 1 and table 2, the authors should include all CI for OR;
2, how many patients have CC or CT genotype? Any MI or PAOD difference among them?
3, the authors should provide full word for abbr at the first time they used, such as PAOD, T1DM...
Author Response
Point-by-point-reply
Reviewer Comments:
Reviewer 1
Comments and Suggestions for Authors
The authors did a retrospective study involving 436 renal allograft recipients, and found that the patients with homozygous GNB3 T allele have lower myocardial infarction and acute peripheral artery occlusive
disease free survival rate.
Overall, the results are generally well presented, but a few issues will need to be addressed within the revised text of the manuscript before the paper can be published.
1, in table 1 and table 2, the authors should include all CI for OR;
Answer to Question 1:
As requested, we added the data on confidence intervals to the odds ratios in Table 1 and 2.
2, how many patients have CC or CT genotype? Any MI or PAOD difference among them?
Answer to Question 2:
Among 393 C-allele carriers, 205 renal allograft recipients had the CC genotype and 188 recipients carried the CT genotype. The incidence of acute myocardial infarction after renal transplant was comparable between carriers of the CT and CC genotype (11 (6%) vs. 16 (8%), p=0,44). Similarly, no difference was observed in appearance of acute PAOD posttransplant comparing CT genotypes with CC genotype carriers (11 (6%) vs. 10 (5%), p=0,67). In terms of myocardial infarction-free survival and acute PAOD-free survival, we detected comparable survival rates for the CT and CC genotype carriers (p=0,33 and p=0,7).
3, the authors should provide full word for abbr at the first time they used, such as PAOD, T1DM...
Answer to Question 3:
As requested, we checked the text of the manuscript for abbreviations and added the full word in the cases, when abbreviation was mentioned for the first time.
Additionally, we provided a list of abbreviations in the revised version of the manuscript.
Reviewer 2 Report
Dear Editor,
This manuscript investigates the
GNB3 c.825C>T (rs5443) polymorphism and risk of acute cardiovascular events after renal allograft transplant. This study reveals that the homozygous GNB3 T allele exerts noticeable effects on the risk of MI and acute PAOD only in the presence of additional nonheritable risk factors.
Overall, the manuscript is very concise and informative to readers, authors reported new information on the effect of the GNB3 c.825C>T polymorphism on cardiovascular events among renal allograft recipients post-transplant.
However, some questions arise from the study and the authors need to address:
1. Authors should further validate c.825C>T polymorphism including data of more patients with liver or lung diseases to see whether polymorphism functions.
2. Authors should provide a model figure based on their results to understand easily to readers.
3. Authors should discuss about other polymorphisms in GNB3 gene if there are any reported.
4. In the introduction authors should elaborate on the cellular functions of G protein and how it controls signal transduction in cells.
5. Other published polymorphisms in genes of β subunit and their effect on acute cardiovascular events after renal allograft transplant if reported any.
6. How truncated Gβ3s with wild-type Gβ3 exhibit a gain of function and subsequent enhanced activation of G protein-induced signaling pathways. Authors should explain in detail, It would be easy to understand.
Minor comment: there are some typo and errors in the text, gene names accordingly, and the resolution of the images is not very high.
